

# Comparative analysis of twelve mitogenomes of Caliscelidae (Hemiptera: Fulgoromorpha) and their phylogenetic implications

Nian Gong[1,2,3], Lin Yang[1,2,3] and Xiangsheng Chen[1,2,3]

[1] Guizhou University, Institute of Entomology, Guiyang, Guizhou, China
[2] Guizhou University, The Provincial Special Key Laboratory for Development and Utilization of Insect Resources, Guiyang, Guizhou, China
[3] Guizhou University, The Provincial Key Laboratory for Agricultural Pest Management of Mountainous Regions, Guiyang, Guizhou, China

## ABSTRACT

Here, the complete mitochondrial genomes (mitogenomes) of 12 Caliscelidae species, *Augilina tetraina, Augilina triaina, Symplana brevistrata, Symplana lii, Neosymplana vittatum, Pseudosymplanella nigrifasciata, Symplanella brevicephala, Symplanella unipuncta, Augilodes binghami, Cylindratus longicephalus, Caliscelis shandongensis,* and *Peltonotellus* sp., were determined and comparatively analyzed. The genomes varied from 15,424 to 16,746 bp in size, comprising 37 mitochondrial genes and an A+T-rich region. The typical gene content and arrangement were similar to those of most Fulgoroidea species. The nucleotide compositions of the mitogenomes were biased toward A/T. All protein-coding genes (PCGs) started with a canonical ATN or GTG codon and ended with TAN or an incomplete stop codon, single T. Among 13 PCGs in 16 reported Caliscelidae mitogenomes, *cox1* and *atp8* showed the lowest and highest nucleotide diversity, respectively. All PCGs evolved under purifying selection, with *atp8* considered a comparatively fast-evolving gene. Phylogenetic relationships were reconstructed based on 13 PCGs in 16 Caliscelidae species and five outgroups using maximum likelihood and Bayesian inference analyses. All species of Caliscelidae formed a steadily monophyletic group with high support. Peltonotellini was present at the basal position of the phylogenetic tree. Augilini was the sister group to Caliscelini and Peltonotellini.

## INTRODUCTION

The family Caliscelidae (Insecta: Hemiptera: Fulgoroidea) includes a diverse group of phytophagous insects, including two subfamilies (Ommatidiotinae and Caliscelinae), five tribes (Caliscelini, Peltonotellini, Ommatidiotini, Adenissini, and Augilini), and >240 species (*Bourgoin, 2021*). The group is relatively small but widely distributed worldwide. Similar to other hemipteran insects, those belonging to the family Caliscelidae use piercing and sucking mouthparts to consume plant juice. Such feeding activities result in proliferation of plant cells which can further affect plant growth and development, spread

Corresponding author
Xiangsheng Chen,
chenxs3218@163.com

plant viral diseases, and severely impact food production. Currently, most research on Caliscelidae is largely focused on the identification and description of new species; however, in a few studies, the partial mtDNA sequences of Caliscelidae species, such as *Bruchomorpha beameri* and *Aphelonema simplex* (*Gwiazdowski et al., 2015*; *Hebert et al., 2016*), have been presented. To date, among 400 hemipteran mitochondrial genomes (mitogenomes) uploaded to GenBank, only four are active caliscelid mitogenomes (*Gong, Yang & Chen, 2021a*). Considering that at least 240 caliscelid species exist, current data on caliscelid planthopper mitogenomes are extremely limited.

Insect mitogenomes are circular double-stranded DNA molecules that are maternally inherited and range from 14 to 36 kb in size (*Cameron, 2014*; *Wang & Tang, 2017*). In metazoans, the typical mitogenome includes 37 genes, containing two ribosomal RNAs (rRNAs: *rrnL* and *rrnS*), 13 protein-coding genes (PCGs) encoding the subunits of oxidative phosphorylation enzymes, 22 transfer RNAs (tRNAs) involved in amino acid transport, and a noncoding A+T-rich region with variable length that serves as the site of initiation for transcription and gene replication (*Wolstenholme, 1992*; *Shadel & Clayton, 1997*; *Inohira, Hara & Matsuura, 1997*; *Boore, 1999*; *Osigus et al., 2013*). Mitogenomes can have genome-level characteristics, such as relatively conserved gene content, base composition, modes of transcription and replication, and gene organization (*Simon et al., 2006*; *Gissi, Iannelli & Pesole, 2008*). Furthermore, they are highly applicable to phylogenetic analyses in higher taxonomic categories, species identification, and population genetics (*Dowton, Castro & Austin, 2002*; *Ingman et al., 2000*; *Ma et al., 2012*).

In the present study, the mitogenomes of 12 species belonging to the family Caliscelidae were determined. All the species were collected from gramineous plants, in which their nymphs and adults are sustained by fresh plant leaves (*Che, Zhang & Webb, 2009*; *Chen, Zhang & Chang, 2014*; *Gong, Yang & Chen, 2018*). To further protect the quantity and quality of gramineous plants, it is necessary to determine the structure and characteristics of the mitogenomes of Caliscelidae, which is likely to provide insights into pest control strategies against these insects (*Tian et al., 2001*). Furthermore, determining the complete mitogenomes of the 12 studied species will enrich the Caliscelidae database for further study of Fulgoroidea phylogenetic relationships and the phylogenetic position of Caliscelidae.

In previous mitogenome-based investigations, only partial genome sequences or limited taxon sampling associated with Fulgoroidea including the Caliscelidae were conducted. Therefore, a comprehensive phylogenetic investigation into the Fulgoroidea superfamily is required based on more taxa and greater mitochondrial sequence coverage. Accordingly, we determined and comparatively analyzed the complete mitogenomes of 12 Caliscelidae species, namely, *Augilina tetraina*, *Augilina triaina*, *Symplana brevistrata*, *Symplana lii*, *Neosymplana vittatum*, *Pseudosymplanella nigrifasciata*, *Symplanella brevicephala*, *Symplanella unipuncta*, *Augilodes binghami*, *Cylindratus longicephalus*, *Caliscelis shandongensis*, and *Peltonotellus* sp., for the first time in the present study. Our analyses specifically included gene order, nucleotide composition, codon usage, tRNA secondary structure, gene overlaps, and noncoding regions. In addition, we conducted a comprehensive phylogenetic investigation on Caliscelidae with expanded mitochondrial

gene data and taxon sampling. Ultimately, our aim was to use all available caliscelid mitogenomes to improve the current understanding of Caliscelidae phylogeny.

## MATERIALS AND METHODS

### Species collection and taxonomic identification

Specimens of *Augilina tetraina* (24°70′N, 97°90′E), *Augilina triaina* (21°58′N, 100°68′E), *Symplana lii* (21°41′N, 101°25′E), *Neosymplana vittatum* (24°69′N, 97°93′E), *Pseudosymplanella nigrifasciata* (21°41′N, 101°25′E), *Symplanella brevicephala* (21°41′N, 101°25′E), *and Augilodes binghami* (21°58′N, 100°68′E) were collected from the bamboo forest of Yunnan province in July 2019, *Symplanella unipuncta* (18°41′N, 109°68′E) was collected from the bamboo forest of Hainan province in May 2021, *Caliscelis shandongensis* (37°58′N, 118°68′E) was collected from phragmites of Shandong province in August 2016, *Peltonotellus* sp. (49°22′N, 119°77′E) was collected from the grassland of Inner Mongolia province in July 2018, and those of *Symplana brevistrata* (25°25′N, 107°75′E) and *Cylindratus longicephalus* (26°21′N, 108°22′E) were collected from Guizhou province in August 2019. All specimens were deposited at the Institute of Entomology, Guizhou University, Guiyang, China, where they were stored in 100% ethanol at −20 °C until further use. These 12 species were identified by Nian Gong according to morphological descriptions and illustrations (especially of genitalia) provided by *Chen, Zhang & Chang (2014)*, *Meng, Gnezdilov & Wang (2015)*, and *Gong, Yang & Chen (2020)*.

### DNA extraction and sequencing

Total genomic DNA was extracted from the muscle tissue of each Caliscelidae species using the Takara Genomic DNA Extraction Kit (Sangon Biotech, Shanghai, China). Use a dissecting needle to remove the muscle tissue of these insects. Specimens were incubated at 56 °C for 6 h to lyse completely and the total genomic DNA was eluted in 50 ml double-distill water (ddH2O), while the remaining steps were following the manufacturer's instructions. Quality of the extracted DNA was checked on 1% agarose gel. The extracted genomic DNA was stored at −20 °C until further use. Voucher specimens with male genitalia and DNA samples have been deposited at the Institute of Entomology, Guizhou University, Guiyang, China. Complete mitogenomes were sequenced at Berry Genomic (Beijing, China). Following quantification of the extracted total genomic DNA, an Illumina TruSeq library for a single species was obtained from the pooled genomic DNA with an average insert size of 350 bp. This library was sequenced on a full run of Illumina Hiseq 2,500 with 500 cycles and paired-end sequencing (150 bp reads).

### Genome assembly, annotation, and analysis

FastQC v0.11.4 (www.bioinformatics.babraham.ac.uk/projects/fastqc) was used to evaluate the quality of the raw sequences; those with an average quality value of <Q30 of the putative mitochondrial genome reads were filtered out before assembly. The clean sequences were then assembled using the MitoZ v2.4 software (*Meng et al., 2019*) with the default parameters.

The 12 mitogenomes were initially annotated using the MITOS web server (http://mitos.bioinf.uni-leipzig.de/index.py) (*Bernt et al., 2013*) with invertebrate genetic codes. The locations and secondary structures of 22 tRNA genes were predicted using the MITOS WebServer and tRNAscan-SE search server (http://lowelab.ucsc.edu/tRNAscan-SE) (*Schattner, Brooks & Lowe, 2005*) with the extended option of invertebrate codon predictors (*Lowe & Eddy, 1997*). Thirteen PCGs were predicted by determining their open reading frames using the invertebrate mitochondrial genetic codons. AT-rich regions and two rRNA genes were determined based on the locations of adjacent tRNA genes and comparisons with homologous genes from other species of Caliscelidae. Mitogenomic circular maps were created and annotated using Geneious R9 (*Kearse et al., 2012*).

The nucleotide composition and relative synonymous codon usage (RSCU) were obtained using PhyloSuite (*Zhang et al., 2020b*), and RSCU figures were created using the fmsb package (*Bivand et al., 2019*) of R 3.6.1 (*R Core Team, 2019*). The composition of skew was calculated according to the following formulas: AT skew = $(A - T)/(A + T)$; GC skew = $(G - C)/(G + C)$ (*Perna & Kocher, 1995*). Repeated sequences in the A+T-rich region were found using the Tandem Repeats Finder program (http://tandem.bu.edu/trf/trf.html) (*Benson, 1999*). The overlapping regions and intergenic spacers between genes were manually counted. The data of nucleotide diversity and the ratio of nonsynonymous substitutions (Ka) to synonymous substitutions (Ks) for all PCGs were collected as previously described in *Yang et al. (2019)*, specifically, calculated using DNASP v5.0 (*Librado & Rozas, 2009*). The sequence data of the 12 insect mitogenomes have been deposited in GenBank under the accession numbers MT577030 for *Neosymplana vittatum*, MW550299–MW550301 for *Symplanella brevicephala*, *Symplana brevistrata*, and *Symplana lii*, MW928525–MW928530 for *Cylindratus longicephalus*, *Augilodes binghami*, *Pseudosymplanella nigrifasciata*, *Augilina triaina*, *Augilina tetraina*, and *Peltonotellus* sp., and MZ343194–MZ343195 for *Caliscelis shandongensis* and *Symplanella unipuncta*, respectively.

## Phylogenetic analyses

In addition to the 12 mitogenomes sequenced in this study, the complete mitogenomes of nine planthopper species were downloaded from GenBank for phylogenetic analyses; the mitogenomes of *Ricania marginalis*, *Ricania speculum*, *Sivaloka damnosus*, *Hemisphaerius rufovarius*, and *Geisha distinctissima* were used as outgroups. Detailed information and accession numbers of these mitogenomes are listed in Table S1.

The nucleotide sequences of 13 PCGs were aligned using MEGA v6 (*Tamura et al., 2013*) with Muscle (*Edgar, 2004*). Individual genes were concatenated using SequenceMatrix v1.7 (*Vaidya, Lohman & Meier, 2011*). The optimal partition strategy and substitution models for Bayesian inference (BI) and maximum likelihood (ML) analyses were selected using the partition scheme of the software PartitionFinder v2.1.1 (*Lanfear et al., 2017*) with the greedy algorithm (*Lanfear et al., 2012*). The best-selected partitioning schemes and models for ML and BI analyses are listed in Table S2. ML analyses were performed using IQ-TREE v1.6.3 (*Nguyen et al., 2014*) with 10,000 replicates of ultrafast likelihood bootstrapping (*Minh, Nquyen & von Haeseler, 2013*) to obtain node support values. BI analyses were conducted using MrBayes v3.2.215 (*Ronquist et al., 2012*) under
the following conditions: two independent Markov chain Monte Carlo runs for 1,000,000 generations, sampling every 1,000 generations, with a burn-in of 25%. The convergence between the two runs was established by Tracer v1.6 (effective sample size > 200) (*Rambaut et al., 2014*). Stationarity was considered to have been reached when the average standard deviation of split frequencies decreased to <0.01 and remained stable. Bootstrap percentages (BPs) of >75% or Bayesian posterior probabilities (BPPs) of >0.9 were considered credible (*Hillis & Bull, 1993*; *Mutanen, Wahlberg & Kaila, 2010*). Finally, phylogenetic trees were viewed and edited using FigTree 1.4.2 (*Mousavi et al., 2014*).

## RESULTS

### Genome organization and composition

The 12 newly sequenced mitogenomes were all circular double-stranded molecules with lengths ranging from 15,424 bp in *Pseudosymplanella nigrifasciata* to 16,746 bp in *Neosymplana vittatum* (Fig. 1; Table S3). Each newly sequenced mitogenome presented 37 typical metazoan mitochondrial genes, including 13 PCGs, 22 putative tRNA genes, two rRNA genes, and a large noncoding A+T-rich region (Fig. 1). Among these 37 genes, 23 (9 PCGs and 14 tRNAs) were found on the major strand (J-strand), whereas the remaining 14 (4 PCGs, 2 rRNAs, and 8 tRNAs) were found on the minor strand (N-strand) (Fig. 1; Table S3).

The AT nucleotide content of the 12 mitogenomes was similar: an average of 78.4% to 79.6% (Table S4). They all showed a positive AT skew (0.119 to 0.227) and a negative GC skew (−0.298 to −0.177), indicating strong AT bias, similar to that in other Caliscelidae insects (*Gong, Yang & Chen, 2021a*). The highest and lowest A+T content was present in the control region (80.2%–89.1%) and PCGs (77.5%–78.8%), respectively (Table S4), similar to that in all previously sequenced mitogenomes of fulgoroid planthoppers (*Xu, Long & Chen, 2019*).

### PCGs and codon usage

The total length of the 13 PCGs of 12 species were ranging from 10,931 to 10,983 bp; their average AT content was 77.5% to 78.8%. The AT (−0.155 to −0.122) and GC (−0.088 to −0.036) skewness of the PCGs was similar among the 12 planthopper species (Table S4). Of the PCGs, the gene length of *nd5* and *atp8* was the longest and shortest, respectively (Table S3). Four of the 13 PCGs (*nd1*, *nd4*, *nd4l*, and *nd5*) were coded on the minor strand, whereas the other nine (*cox1*, *cox2*, *cox3*, *atp6*, *atp8*, *nd2*, *nd3*, *nd6*, and *cytb*) were encoded on the major strand (Table S3; Fig. 1).

In the 12 newly sequenced mitogenomes, most PCGs used the typical ATN as initiation codons. However, the start codon GTG was also used in *nd1* and *nd5*. The typical stop codon TAA occurred more frequently than TAG, and a single T was also frequently used as the stop codon. The presence of an incomplete stop codon is common in insects; it is believed to be completed by posttranscriptional polyadenylation (*Ojala, Montoya & Attardi, 1981*).

The codon usage pattern, RSCU, and number in the PCGs of the 12 planthopper mitogenomes were determined (Figs. 2 and 3; Table S5). Analysis of PCG codon usage

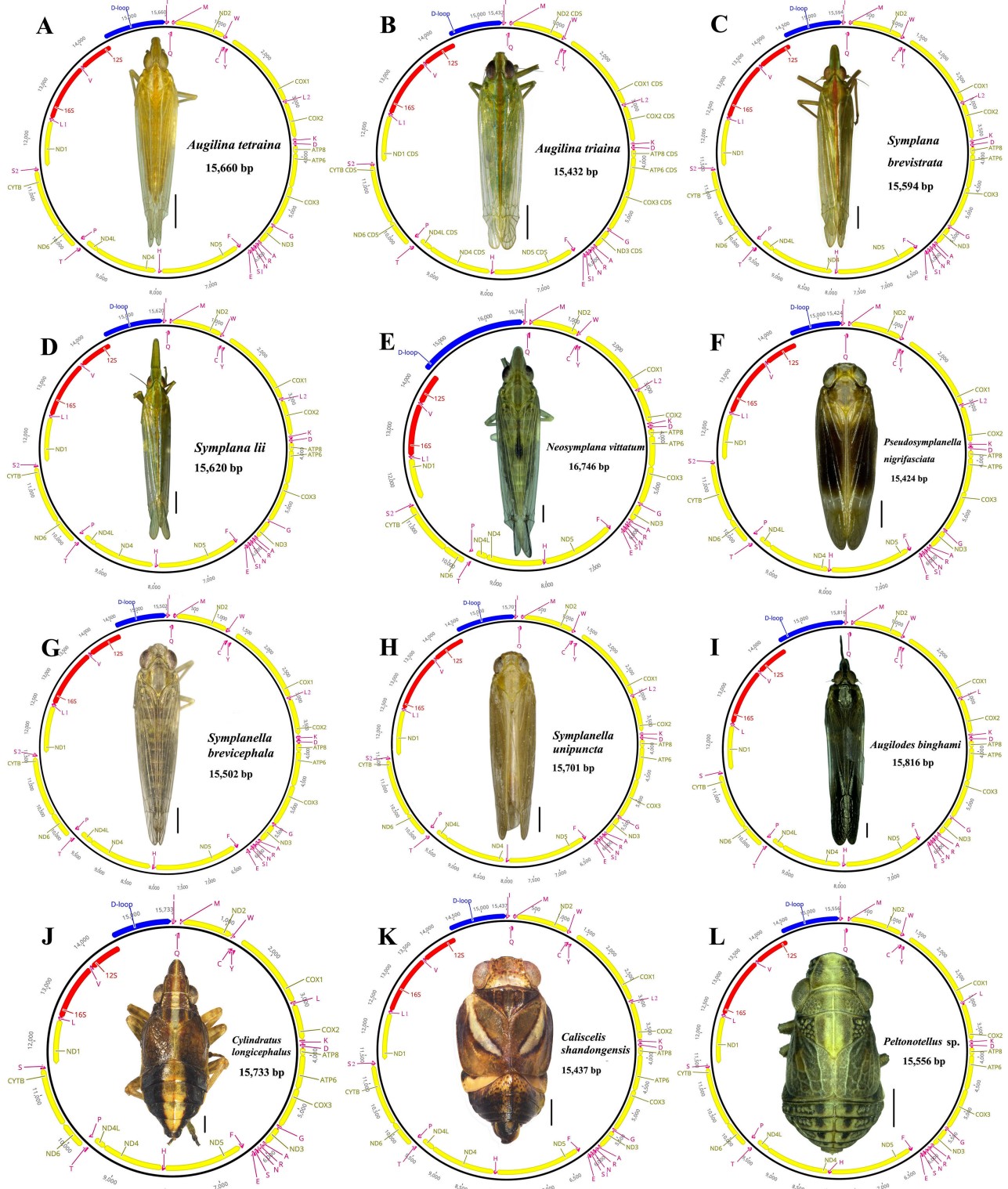

**Figure 1 Circular maps of the 12 newly sequenced mitogenomes of Caliscelida3 (A–L).** Protein-coding, ribosomal, and transfer RNA genes are shown with standard abbreviations. Gene orientations are indicated by arrow directions. Protein-coding genes, transfer RNA genes, control regions, and two ribosomal RNA genes are shown in yellow, aubergine, blue, and red, respectively.

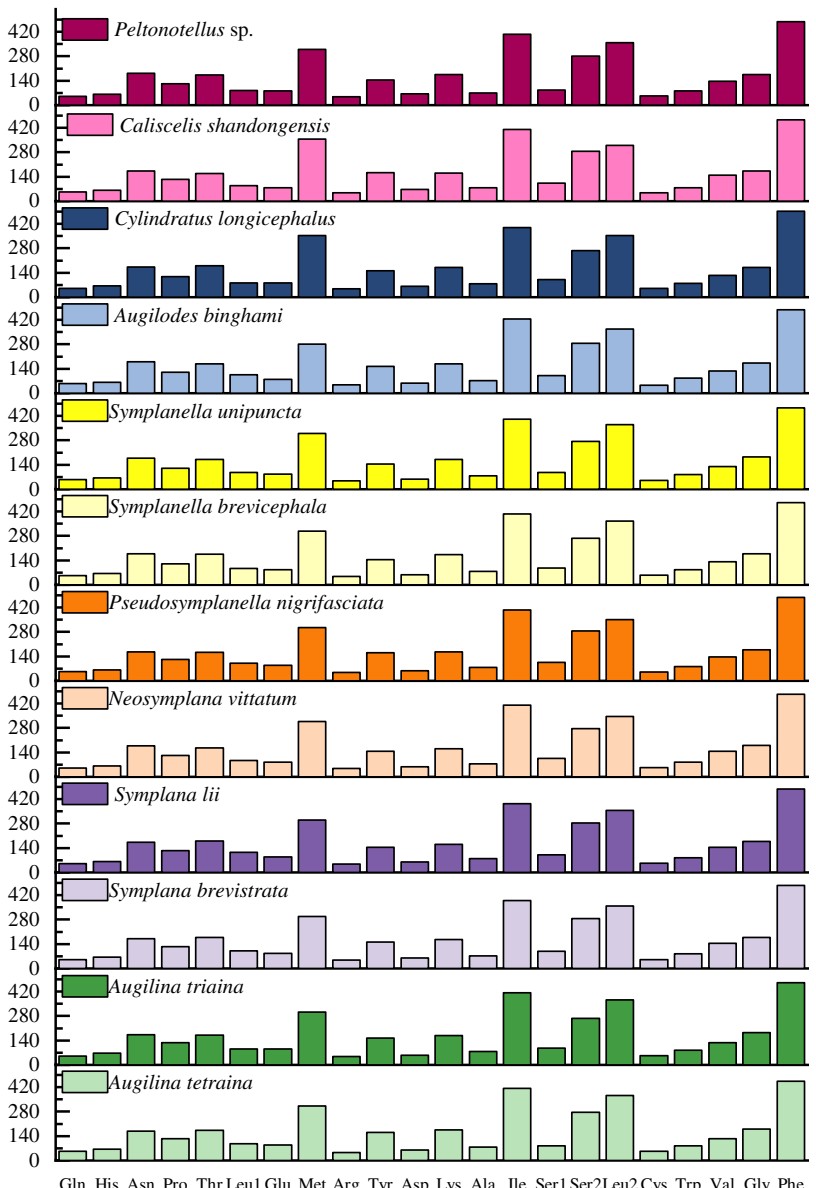

**Figure 2 Number of codon usages in the protein-coding genes of the 12 newly sequenced mitogenomes of Caliscelidae.**

showed similar results among species, with Phe-UUU, Ile-AUU, Leu2-UUA, and Met-AUA being most frequently used as codons (Fig. 2); these accounted for >50% of the total number of amino acids. All were solely composed of A or U, which is reflected in the high A+T content of the PCGs. *Augilina triaina* included 59 available codons, *Symplanella unipuncta* included 60 available codons, *Symplana brevistrata* included 61 available codons, *Augilina tetraina*, *Symplanella brevicephala*, *Augilodes binghami*, *Cylindratus longicephalus*, *Caliscelis shandongensis*, and *Peltonotellus* sp. included 62 available codons, *Symplana lii* and *Neosymplana vittatum* included 63 available codons, whereas

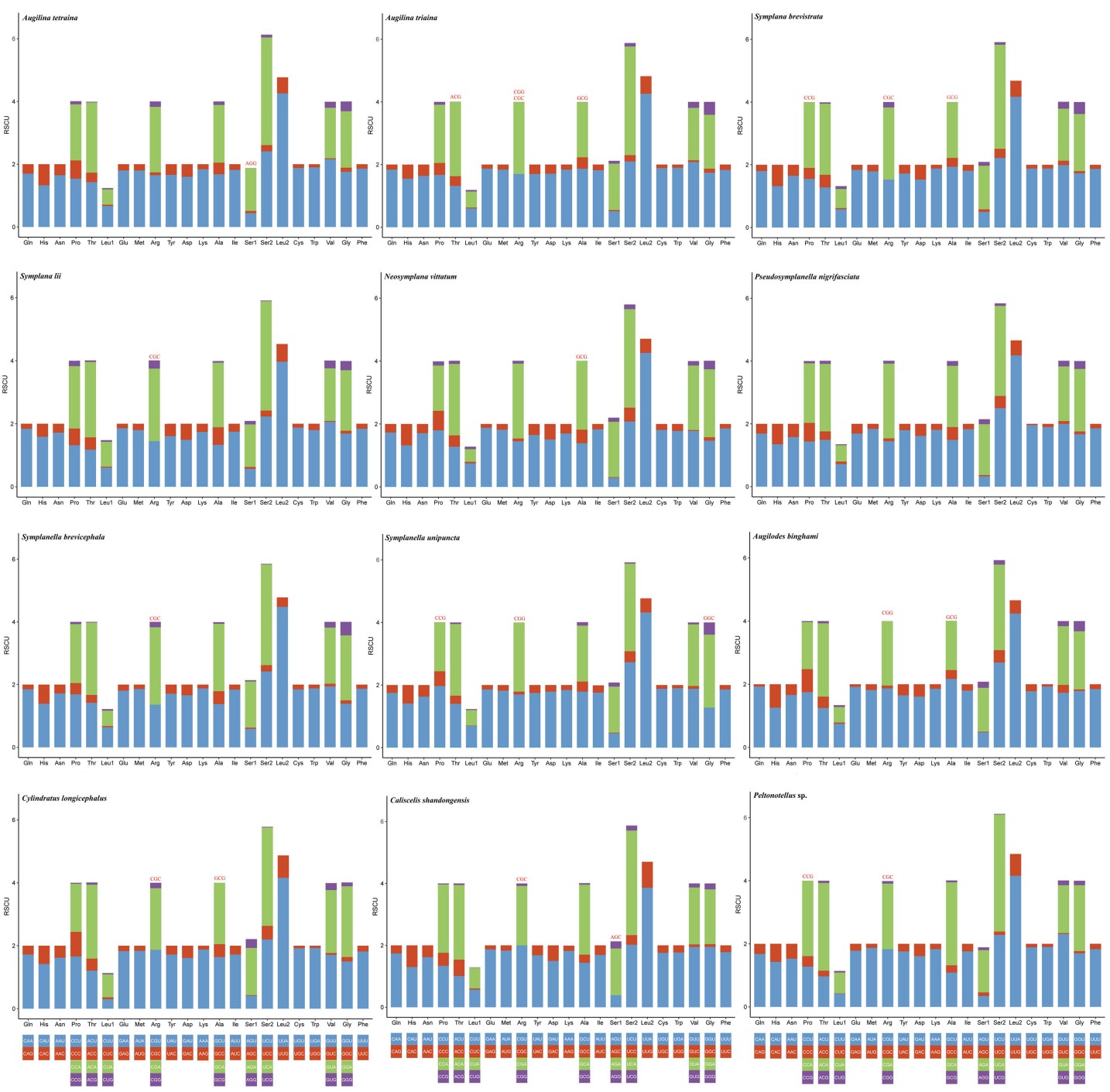

**Figure 3 Relative synonymous codon usage in the protein-coding genes of the 12 newly sequenced mitogenomes of Caliscelidae.** Codon families are indicated below the X-axis. The color of the codon family below the X-axis corresponds to the color above the X-axis. The stop codon is not given. Codons absent in mitogenomes are shown at the top of columns.

*Pseudosymplanella nigrifasciata* included 64 available codons (excluding TAA and TAG) (Table S5). The RSCU values of the PCGs showed a trend toward the use of A/T compared with that of G/C. Furthermore, codon usage revealed an extremely high A+T bias that

played a pivotal role in the A+T bias of the complete mitogenome. Among the 12 newly sequenced mitogenomes, the AT and GC skews of the PCGs were negative.

## rRNA and tRNA genes

Both small rRNA (*rrnS*) and large rRNA (*rrnL*) genes were found in all of the newly sequenced mitogenomes; they were located between *trnV* and the A+T-rich region and between *trnV* and *trnL1*, respectively, and both were oriented on the minor strand (Fig. 1). Among the 12 mitogenomes, the length of *rrnL* ranged from 1,205 bp in *Caliscelis shandongensis* to 1,238 bp in *Augilodes binghami*; the length of *rrnS* ranged from 709 bp in *Pseudosymplanella nigrifasciata*, *Symplanella brevicephala*, and *Symplanella unipuncta* to 732 bp in *Caliscelis shandongensis*. For the rRNAs, the AT-skew was negative (−0.21 to −0.129) and GC-skew was positive (0.256 to 0.354); these genes had the second highest A+T content in the genome (Table S4).

There was a typical set of 22 tRNAs sized 55–74 bp and interspersed throughout the mitogenome (Fig. 1; Table S3). The overall length of all tRNAs was in the range of 1,409–1,455 bp. Among these genes, 14 and 8 tRNAs were encoded by the major and minor strands, respectively. For all tRNAs, the AT-skew and GC-skew were positive and the A+T content was high (>77.8%; Table S4). Of the 22 tRNAs, two genes (*trnS1*, and *trnV*) of *Cylindratus longicephalus*, *Caliscelis shandongensis*, and *Peltonotellus* sp. of the subfamily Caliscelinae lacked a DHU stem, and three genes (*trnC*, *trnS1*, and *trnV*) of the remaining nine species of the subfamily Ommatidiotinae lacked a dihydrouridine arm (DHU stem) and therefore formed a simple loop, whereas the remaining could fold into canonical cloverleaf secondary structures (Figs. S1–S12).

Based on the alignment of tRNAs of 12 Caliscelidae species, the percentage of identical nucleotides (%INUC) was calculated (Table S6). The *trnK* located on the major strand had the highest %INUC (79.45%), whereas *trnC* on the minor strand had a %INUC of only 33.78%. Three tRNAs (*trnE*, *trnK*, and *trnL2*) located on the major strand exhibited high levels of conservation with an average identity of >70%, indicating that tRNAs on the major strand were highly conserved. The tRNA stem was highly conserved. The base variation of the anticodon loops was low and relatively conserved, and the size of the anticodon loops was highly conserved (all 7 bp); the remaining loops showed different degrees of base variation, and the TψC loops showed the most variation. Thus, the nucleotide substitutions occurred less on tRNA stems than on loops, indicating higher conservation of tRNA stems.

## Overlapping and intergenic spacer regions

The 12 mitogenomes contained 13–23 intergenic spacers, which ranged in size from 1 to 66 bp. *Augilina tetraina* had the longest intergenic spacer (66 bp), situated between *trnH* and *nad4*. The 12 species had 5–11 overlapping genes, with overlaps ranging in size from 1 to 8 bp (Table S3). Three gene overlaps were conserved among the 12 newly sequenced mitogenomes: *trnW*–*trnC* (8 bp: AAGCCTTA), *atp8*–*atp6* (7 bp: ATGATAA) except *Caliscelis shandongensis*, and *nad4*–*nad4L* (7 bp: TTAACAT).

## The AT-rich region

The AT-rich region is involved in regulating the replication and transcription of the mitogenome in insects (*Zhang & Hewitt, 1997*). The control region is the largest noncoding region present between *rrnS* and *trnI* (Fig. 1). Among the 12 species, the length of the AT-rich regions ranged from 980 bp in *Symplanella brevicephala* to 2,319 bp in *Neosymplana vittatum*. The AT-rich region contained the highest AT content of the complete mitogenome, ranging from 80.2% in *Peltonotellus* sp. to 89.1% in *Symplanella unipuncta*. The AT skewness of *Augilodes binghami*, *Symplanella unipuncta*, *Symplanella brevicephala*, and *Symplana brevistrata* was negative, indicating a pattern toward T compared with A. All species had a negative GC skew (−0.464 to −0.074). The structural organization of the AT-rich regions in the 12 planthopper mitogenomes is illustrated in Figs S13–S24. Four repeat regions were present in *Neosymplana vittatum*, two were present in *Peltonotellus* sp., *Cylindratus longicephalus*, *Augilodes binghami*, *Pseudosymplanella nigrifasciata*, *Symplanella unipuncta*, *Symplanella brevicephala*, *Symplana lii*, and *Symplana brevistrata*, whereas one repeat region was present in *Caliscelis shandongensis*, *Augilina tetraina*, and *Augilina triaina*. The largest repeat unit, with two repeats in *Augilina triaina*, was 186 bp in length.

## Phylogenetic relationships

All 21 species from Fulgoroidea (including the 12 species with newly sequenced mitogenomes, four Caliscelidae species were downloaded from GenBank and five outgroup species) were subjected to phylogenetic analysis based on the concatenated nucleotide sequences of 13 PCGs; using ML and BI analyses, two phylogenetic trees were obtained. These two trees had the most consistent topologies and higher node support values; thus, they were merged (Fig. 4). In the consent tree, all species of Caliscelidae formed a steadily monophyletic group with high support (BPPs = 1; BPs = 100). At the genus level, (*Peltonotellus*+ {*Caliscelis* + [*Cylindratus* + *Bambusicaliscelis*]}) formed one clade with high support (BPPs = 1; BPs = 100), while (*Youtuus* + {*Augilodes* + [(*Symplanella* + <*Pseudosymplanella* + *Neosymplana*>) + <*Symplana* + *Augilina*>]}) formed one clade with high support (BPPs = 1; BPs > 99). At the tribe level, Augilini formed sister group with the clade containing Peltonotellini and Caliscelini, Caliscelini was the sister group to Peltonotellini. At the subfamily level, Caliscelinae was the sister group to Ommatidiotinae with high nodal support (BPPs = 1; BPs = 100).

## Nucleotide diversity and evolutionary rate analysis

To investigate the evolutionary rates of the mitochondrial PCGs, nucleotide diversity, Ka, Ks, and the Ka/Ks ratio were calculated across 16 mitogenomes of Caliscelidae for each aligned PCG (Fig. 5; Table S7) (*Nei & Gojobori, 1986*). Nucleotide diversity values for the individual genes ranged from 0.164 (*cox1*) to 0.338 (*atp8*). Other genes with comparatively low nucleotide diversity values included *cox1* (0.164), *nad1* (0.177) and *cytb* (0.177), whereas those with comparatively high values included *atp8* (0.338), *nad6* (0.271), and *nad2* (0.266). The Ka/Ks substitution ratio can be used to estimate whether a sequence is undergoing purifying, neutral, or positive selection. In pairwise Ka/Ks analyses of the 16

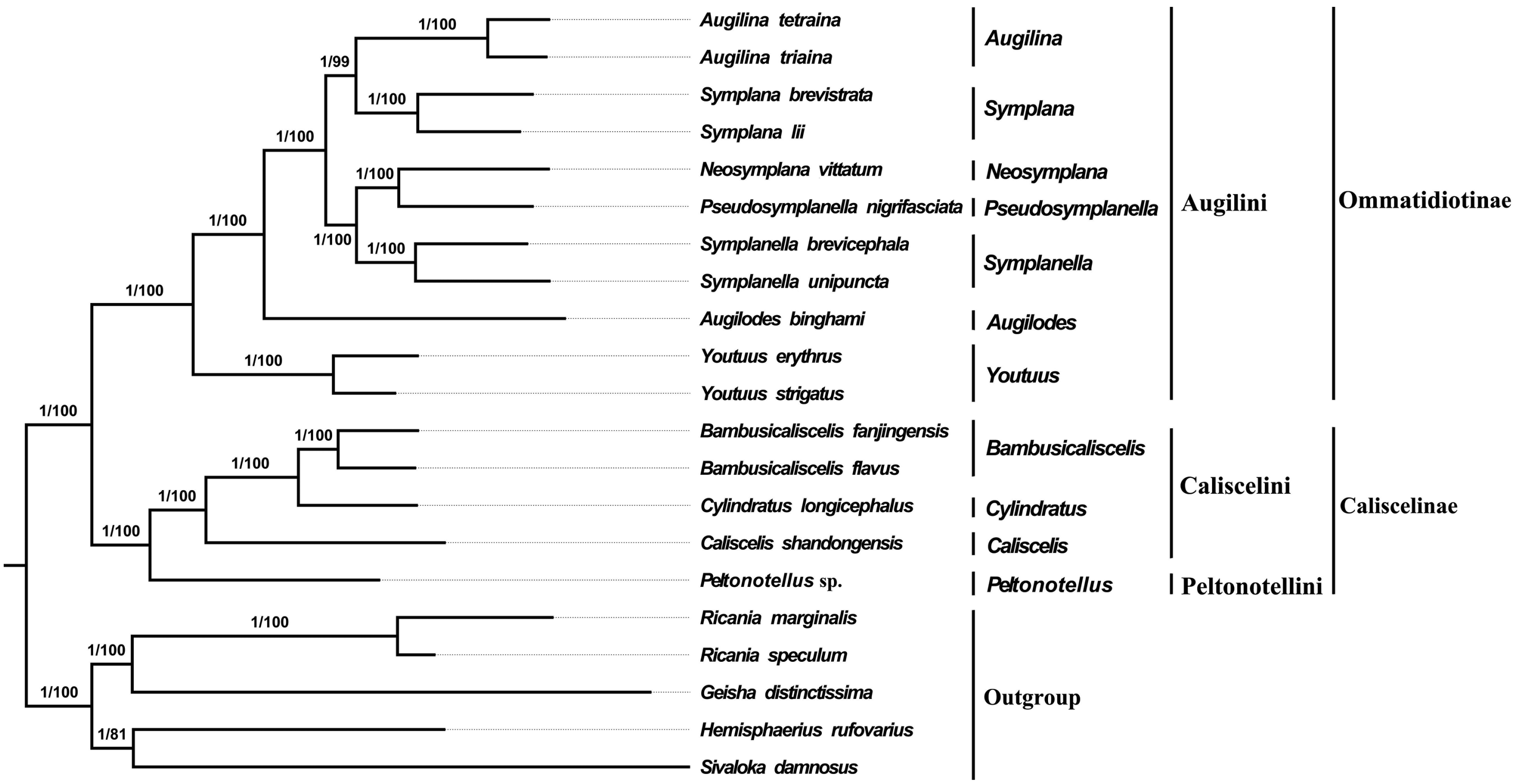

**Figure 4 Phylogenetic trees of Fulgoroidea inferred using MrBayes (Bayesian inference) and maximum likelihood (ML) analysis based on the nucleotide sequences of 13 protein-coding genes.** Bayesian posterior probabilities (BPPs) and bootstrap percentages (BPs) are indicated on branches.

sequenced mitogenomes, average Ka/Ks ratios ranged from 0.093 for *cox1* to 0.735 for *atp8*; all ratios were consistently < 1, indicating that all PCGs from the Caliscelidae mitogenomes were evolving under purifying selection. From these analyses, it can be concluded that *cox1* exhibited the strongest purifying selection, whereas the nad family genes exhibited a slightly relaxed purifying selection. In addition, *atp8* exhibited the least selection pressure and the fastest evolution rate among the mitochondrial PCGs of Caliscelidae, consistent with the findings of previous research (*Yang et al., 2019*).

## DISCUSSION

Using next-generation sequencing, we successfully sequenced and then analyzed 12 complete mitochondrial genomes of species from Caliscelidae. To the best of our knowledge, these mitogenomes add to only four other complete mitochondrial gene sequences in the NCBI database, which does not sufficiently represent >240 known species of Caliscelidae. Compared with the lengths of the four previously sequenced caliscelid sequences (15,922 bp in *Youtuus erythrus* to 16,640 bp in *Youtuus strigatus*) (*Gong, Yang & Chen, 2021a*), the 12 sequences obtained here did not show high variation in length (15,424–16,746 bp). However, the sequences reported here lie between the 14,367 and 17,619 bp of *Nilaparvata lugens* from other known Fulgoroidea mitogenomes (*Zhang*

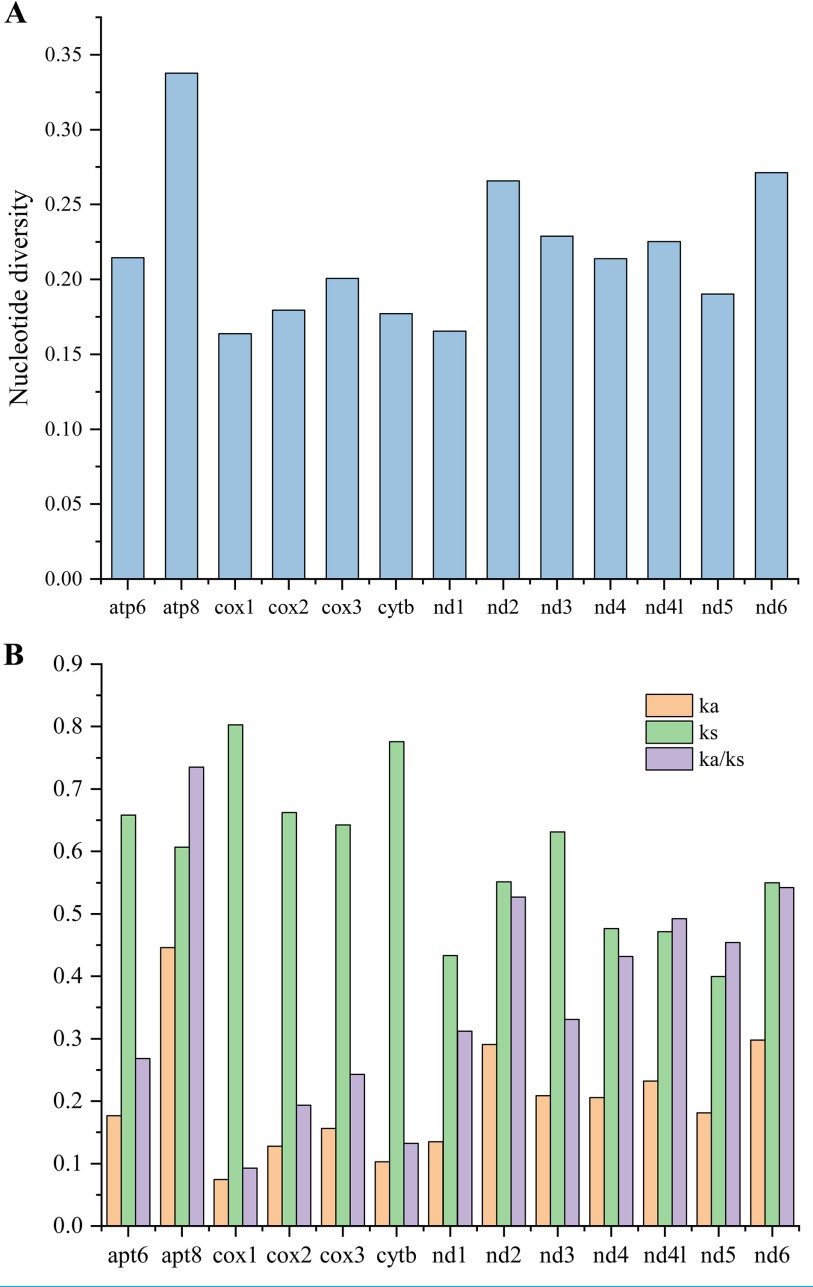

**Figure 5 Nucleotide diversity (A) and the ratio of Ka/Ks (B) of protein-coding genes from 16 reported Caliscelidae mitogenomes.**

*et al., 2013*; *Lv et al., 2015*). These differences in mitogenome size can mainly be attributed to variations in the A+T-rich region and more tandem repeats (*Wang et al., 2019b*). Previously, the presence of gene rearrangement has only been reported in Delphacidae mitogenomes among all the Fulgoroidea insects reported to date; compared with the putative ancestral gene order, the positions of five tRNA genes (*trnC*, *trnW*, *trnH*, *trnP*, and *trnT*) and three PCGs (*nad4*, *nad4l*, and *nad6*) are inverted or translocated (*Zhang et al., 2013*; *Lv et al., 2015*; *Song & Liang, 2009*). However, none of the 12 newly sequenced

caliscelid mitogenomes had any gene rearrangement. Indeed, the gene order and orientation of the mitogenomes of the 12 planthoppers were identical to those of the published mitogenomes of Caliscelidae, consistent with ancestral arthropod mitogenomes (*Boore, 1999*; *Xu, Long & Chen, 2019*; *Huang & Qin, 2018*). Except for *nad6* (29 bp) and *cytb* (21 bp), gene length differences did not exceed 20 bp; *nad2* was of the exact same length, indicating that the PCGs have relatively conserved features. Furthermore, these results are consistent with those from a previous study (*Xu, Long & Chen, 2019*). The location, length, and AT content of rRNAs were highly conserved in the Caliscelidae species sequenced here, similar to those of previously sequenced planthoppers (*Gong, Yang & Chen, 2021a*; *Xu, Long & Chen, 2019*; *Song & Liang, 2009*).

Among the 22 putative tRNA secondary structures, two genes (*trnS1*, and *trnV*) of three species of the subfamily Calisceelinae lacked a DHU stem and three genes (*trnC*, *trnS1*, and *trnV*) of remaining nine of subfamily Ommatidiotinae lacked a DHU stem, and therefore formed a simple loop. The DHU arm of *trnS1* also formed a simple loop, similar to that in several other hemipterans (*Song & Liang, 2009*; *Lee et al., 2009*; *Hua et al., 2009*; *Li et al., 2011*) and metazoans (*Lavrov, Brown & Boore, 2000*). In addition, the lack of a DHU stem in *trnV* has been reported in Caliscelidae as well as in *Aphaena discolor* and *A. amabilis* of Fulgoridae (*Wang et al., 2019a*). The DHU arm of *trnC* formed a simple loop, which has also been reported in Ommatidiotinae of Caliscelidae, whereas *trnC* in Calisceelinae of Caliscelidae is normal (*Gong, Yang & Chen, 2021a*); these differences are species-specific. Furthermore, *trnG* and *trnS2* in *Sogatella furcifera* and *trnH* in *Laodelphax striatellus* of Delphacidae cannot form a typical cloverleaf structure (*Song & Liang, 2009*; *Zhang et al., 2014*; *Yu & Liang, 2018*). In the present study, the percentage of identical nucleotides showed that conservation of tRNAs was high on the major strand, which is consistent with previous findings (*Zhang et al., 2019*). The anticodon arm and loops were the most conserved regions of the tRNAs, but the tRNA stems were always more conserved than the loops, which is in agreement with the findings of previous studies (*Xu, Long & Chen, 2019*; *Zhang et al., 2019*).

In several Fulgoroidea species, an 8-bp overlap of "AAGCCTTA" was detected between *trnW* and *trnC*. This characteristic is consistent with other Caliscelidae species (*Gong, Yang & Chen, 2021a*; *Xu, Long & Chen, 2019*). In addition, we detected a 7-bp overlap "ATGATAA" between *atp8* and *atp6* in the 11 new sequenced studied species; however, in the reported mitogenomes of *Bambusicaliscelis*, the overlap is "ATAATAA". Therefore, perhaps owing to differences within the subfamily, species-specific A/G differences exist. A 7-bp overlap "TTAACAT" was previously found between *nd4* and *nd4l* in Caliscelidae, but it was not found in other Fulgoroidea mitogenomes (*Gong, Yang & Chen, 2021a*; *Xu, Long & Chen, 2019*).

In this study, a total of 11 representative genera of the Chinese Caliscelidae family were selected for phylogenetic analysis to understand the genetic relationship among each genus. The phylogenetic trees generated by BI and ML methods were fully resolved with identical topologies (Fig. 4). Among them, *Peltonotellus* (Peltonotellini) is located at the middle of the evolutionary tree with three genera (*Bambusicaliscelis*, *Cylindratus*, and *Caliscelis*) of the tribe Caliscelini, these two tribes are grouped into the subfamily

Caliscelinae, indicating a relationship similar to the previous results obtained based on morphology. Emeljanov erected the tribe Peltonotellini and the recent modern classification of the family Caliscelidae, including the tribe Peltonotellini was published by Gnezdilov (*Emeljanov, 2008*; *Gnezdilov & Wilson, 2011*; *Gnezdilov, 2013*). Augilini formed sister-group with the clade containing Peltonotellini and Caliscelini, *Youtuus* was basal to the Augilini. *Augilina* and *Symplana* are sister groups, which are so similar morphologically that they can easily be mistaken for the same genus and can only be identified by their genitalia (*Zhang et al., 2020a*), phylogenetic analysis of the mitochondrial genomes of these two genera is consistent with the conclusion of morphological studies by *Gong, Yang & Chen (2021b)* indicating that *Augilina* and *Symplana* should be two genera rather than the same genus. At present, the monophyly of Caliscelidae is yet to be tested because of the limited number of available mitogenomes; currently, mitochondrial data does not exist for Ommatidiotini, and Adenissini in the NCBI database. Thus, more detailed investigation is required to test the monophyly of Caliscelidae. In addition, increasing the number of mitogenome datasets might improve the reliability and robustness of phylogenetic analyses for Fulgoroidea.

The complete mitogenomes of the 12 planthoppers presented here along with the reported phylogenetic relationships may serve as a baseline dataset for future studies on population genetics. Moreover, our study improves the current understanding of mitogenome structure to develop effective diagnostic and management strategies for insect pests.

## CONCLUSIONS

In the present study, we determined and comparatively analyzed the complete mitogenomes of 12 Caliscelidae species, namely, *Augilina tetraina*, *Augilina triaina*, *Symplana brevistrata*, *Symplana lii*, *Neosymplana vittatum*, *Pseudosymplanella nigrifasciata*, *Symplanella brevicephala*, *Symplanella unipuncta*, *Augilodes binghami*, *Cylindratus longicephalus*, *Caliscelis shandongensis*, and *Peltonotellus* sp., for the first time. The 12 mitogenomes ranged from 15,424 to 16,746 bp in length, with the typical gene content and similar arrangement of genes usually observed in Hexapods. Among 13 PCGs in 16 reported Caliscelidae mitogenomes, *cox1* and *atp8* showed the lowest and highest nucleotide diversity, respectively. All PCGs evolved under purifying selection, with *atp8* considered a comparatively fast-evolving gene. The complete mitogenomes of the 12 planthoppers presented here along with the reported phylogenetic relationships may serve as a baseline dataset for future studies on population genetics. Moreover, our study improves the current understanding of mitogenome structure to develop effective diagnostic and management strategies for insect pests.

### Funding

This work was supported by the National Natural Science Foundation of China (grant nos. 32060343, 31472033); the Science and Technology Support Program of Guizhou Province

(grant no. 20201Y129); the Program of Excellent Innovation Talents, Guizhou Province (grant no. 20154021). The funders had no role in study design, data collection and analysis, decision to publish, or preparation of the manuscript.

### Grant Disclosures
The following grant information was disclosed by the authors:
National Natural Science Foundation of China: 32060343, 31472033.
Science and Technology Support Program of Guizhou Province: 20201Y129.
Program of Excellent Innovation Talents, Guizhou Province: 20154021.

### Competing Interests
The authors declare that they have no competing interests.

### Author Contributions
- Nian Gong conceived and designed the experiments, performed the experiments, analyzed the data, prepared figures and/or tables, authored or reviewed drafts of the paper, and approved the final draft.
- Lin Yang performed the experiments, analyzed the data, prepared figures and/or tables, and approved the final draft.
- Xiangsheng Chen conceived and designed the experiments, authored or reviewed drafts of the paper, and approved the final draft.

### DNA Deposition
The following information was supplied regarding the deposition of DNA sequences:
The sequence data of the 12 insect mitogenomes are available in GenBank: MT577030 for *Neosymplana vittatum*, MW550299–MW550301 for *Symplanella brevicephala*, *Symplana brevistrata*, and *Symplana lii*, MW928525–MW928530 for *Cylindratus longicephalus*, *Augilodes binghami*, *Pseudosymplanella nigrifasciata*, *Augilina triaina*, *Augilina tetraina*, and *Peltonotellus* sp., and MZ343194–MZ343195 for *Caliscelis shandongensis* and *Symplanella unipuncta*, respectively.

### Data Availability
The raw measurements are available in the Supplementary Files.

### Supplemental Information
Supplemental information for this article can be found online at http://dx.doi.org/10.7717/peerj.12465#supplemental-information.

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
