# Peer review of "Comparative analysis of twelve mitogenomes of Caliscelidae (Hemiptera: Fulgoromorpha) and their phylogenetic implications"

_PeerJ, doi:10.7717/peerj.12465_

## Round 0.1 · original submission · Major Revisions

Dear authors, please kindly attend to the comments raised by the reviewers. Please work diligently on them.
Thank you very much

·

Basic reporting

L. 83: How does mtDNA contribute to the control of pests?
L.104: It would be better if the authors can provide the information for longitude and latitude.
L.149: PhyloSuite has been published in Molecular Ecology Resources, please cite the up-to-date citation, not the Biorxiv preprint.
L. 180: 1000 ultrafast bootstraps is too few. The authors should use at least 10,000, but you may also use 100,000.
L. 216: Please italicize gene names throughout the MS.

Experimental design

L.174: Did the authors directly align the nucleotide sequences of PCGs without considering the codon frame? PCGs should be aligned in codon mode; you may either use PhyloSuite or MEGA for that.

Validity of the findings

L.240-242: Confirm this, normally AT skew of insects is positive.
L.251-L.252: Please also confirm the skew here.
L. 311: 100 is the max value of the bootstrap support. What is the meaning of “BPs>100” here?
L. 315-318: Peltonotellini was not the basal lineage.
L. 318-319: Augilini doesn’t form a sister group with Caliscelini and Peltonotellini.
L. 422-430: The two clades that the authors described as “top” and “bottom” of the tree respectively were sister-clades, so describing them as top and bottom is wrong.
L.435-436: In the phylogenetic analysis, the authors only included members of this family (except for outgroups), so they can’t conclude this.

Additional comments

The authors present 12 new mitogenomes for Caliscelidae species, the data themselves contribute much to the phylogeny or molecular studies for these insect pests. I only have few concerns regarding the M&M and the interpretation of the phylogenetic tree.

I cannot see Figure 2 anywhere.

Figure 4: replace Ommatidiotini with Ommatidiotinae?

Please note that the line numbers mentioned here are referring to the track changes file.

Reviewer 2 ·

Basic reporting

no comment

Experimental design

No comment

Validity of the findings

No coment

Additional comments

1. When authors start the write the result part, they should clear that how many mitogenome studied in this study. How many species dataset prepared for the phylogenetic study.
Studied mitogenomes belong to which family, subfamily and so…..on
Provide the details of mitogenomes which were retrieved from gene bank.
Why authors have used 5 outgroups. Is there any relevance in this study? Explain it

Authors generated 12 mitogenomes data of family Caliscelidae under two subfamilies Ommatidiotinae (9) Caliscelinae (3).

Supplementary files names (sheets) are not corresponded with text.

Authors has made a confusion that theyy have used 25 mitogenomes or 21 mitogenomes. Somewhere they have mentioned 25 line or in phylogeny 21

Insects are a good model to study the gene arrangement pattern. Here, authors did not discuss the gene arrangement scenario in this family. All the species have conserved gene order or they have some gene arrangement. One paragraph can be added. Authors should compare the Caliscelidae gene order to ancestor.

Author have generated the mitogenome data of unidentified species Peltonotellus sp.. What is the relevance?

Annotated reviews are not available for download in order to protect the identity of reviewers who chose to remain anonymous.

Reviewer 3 ·

Basic reporting

The language is clear and unambiguous. I have no issues with the reporting.

Experimental design

2. Experimental Design
Between Specimens’ collection and DNA extraction, there was no specific description of sample preparation.

This according to the authors, was given attention. However, it doesn't appear like anything was done. I believe to encourage reproducibility, the steps and procedure observed to prepare the sample for use in DNA extraction should be described.
Questions like how the muscle tissue of each Caliscelidae species was collected and prepared for extraction should be answered.

Validity of the findings

Findings appear to agree with the research hypothesis, together with the data provided. Issue with discussions and conclusion were added to the attached reviewer's response document

Additional comments

Additional comments have been added to the attached reviewer's response document.

Annotated reviews are not available for download in order to protect the identity of reviewers who chose to remain anonymous.

---

## Round 0.2 · Minor Revisions

Please authors, kindly find attached reviewer comments, and revise accordingly. Thank you for your patience and understanding. Best regards

·

Basic reporting

no comment

Experimental design

no comment

Validity of the findings

The authors have resolved most of my concerns, however, there still remain some mistakes and some places need further clarification:

1. If the authors didn’t align PCGs in the codon mode, it is a drawback with respect to their phylogenetic analysis. I suggest that authors should re-align PCGs in the codon mode, and construct the phylogenetic tree anew.

2. Regarding the skew, please make sure that you didn’t confuse the two strands.

3. The authors still seem to have misunderstood several parts of the phylogenetic tree. A. You can’t say that Peltonotellini was the basal lineage of the phylogenetic tree, even disregarding the outgroups. B. Augilini formed sister-group with the clade containing Peltonotellini and Caliscelini, so they exhibit equal position. Claiming that Augilini is in the upper part of the tree is wrong. C. Claiming that Yotuus is located in the lower part of the Augilini clade is incorrect. You can say that Yotuus was basal to the Augilini.

Additional comments

no comment

Reviewer 2 ·

Basic reporting

No comments

Experimental design

No comments

Validity of the findings

No comments

Additional comments

No comments

---

## Round 0.3 · accepted · Accept

I am very satisfied with the revised manuscript and appreciates the effort the authors have made. The revised manuscript can now be accepted for publication. Thanks to the peer-review process, the authors were able to improve their work.

Thank you authors for finding PeerJ as your journal of choice, and look forward to your future scholarly contributions. Congratulations and very best regards

·

Basic reporting

The authors present 12 new Caliscelidae mitogenomes and conduct phylogenetic analysis using 16 Caliscelidae species. They resolved most of my concerns, I commend the authors’ effort to obtain and sequence as many species as they could (12) as opposed to sequencing one mitogenome at a time. The results and discussions are generally sound now, and I would like to recommend this manuscript for publication in PeerJ after resolving the following two minor problems:
1. line 364: Peltonotellus (Peltonotellini) is not located at the middle of the tree, imaging that if you rotate the node containing Calisselinae and Ommatidiotinae, Peltonotellus can locate at the top of the tree.
2. line 377: delete ";" after the word mitogenomes.

Experimental design

no comment

Validity of the findings

no comment

Additional comments

no comment